# Multi-modal adaptive gated mechanism for visual question answering

**Yangshuyi Xu** [ORCID] ☯, **Lin Zhang** [ORCID] ☯*, **Xiang Shen**

College of Information Engineering, Shanghai Maritime University, Shanghai, China

☯ These authors contributed equally to this work.
\* linzhang@shmtu.edu.cn

## Abstract

Visual Question Answering (VQA) is a multimodal task that uses natural language to ask and answer questions based on image content. For multimodal tasks, obtaining accurate modality feature information is crucial. The existing researches on the visual question answering model mainly start from the perspective of attention mechanism and multimodal fusion, which will tend to ignore the impact of modal interaction learning and the introduction of noise information in the process of modal fusion on the overall performance of the model. This paper proposes a novel and efficient multimodal adaptive gated mechanism model, MAGM. The model adds an adaptive gate mechanism to the intra- and inter-modality learning and the modal fusion process. This model can effectively filter irrelevant noise information, obtain fine-grained modal features, and improve the ability of the model to adaptively control the contribution of the two modal features to the predicted answer. In intra- and inter-modality learning modules, the self-attention gated and self-guided-attention gated units are designed to filter text and image features' noise information effectively. In modal fusion module, the adaptive gated modal feature fusion structure is designed to obtain fine-grained modal features and improve the accuracy of the model in answering questions. Quantitative and qualitative experiments on the two VQA task benchmark datasets, VQA 2.0 and GQA, proved that the method in this paper is superior to the existing methods. The MAGM model has an overall accuracy of 71.30% on the VQA 2.0 dataset and an overall accuracy of 57.57% on the GQA dataset.

## 1. Introduction

There is countless information in the real world, and people live in an environment where multiple modal information blend [1]. With the development of artificial intelligence, people continue to explore the possibility and feasibility of computer simulation of various multi-modal interaction methods of human beings. In recent years, research on multimodal tasks has gradually emerged, such as Image Caption [2, 3], Visual Dialog [4–6], Visual Common-sense Reasoning (VCR) [7–9], Visual Question Answering (VQA) [10, 11], etc.Compared with single-modal tasks, multi-modal tasks require the model to extract and understand information from a single modality and fuse information from two or more modalities to complete complex reasoning and understanding tasks.

**Data Availability Statement:** The VQA2.0 and GQA datasets employed in the manuscript's experiments are publicly available benchmark datasets for visual question answering tasks. The URL for the VQA2.0 dataset is: https://visualqa.org/download.html and the URL for the GQA dataset is:

https://cs.stanford.edu/people/dorarad/gqa/download.html.

**Funding:** This work was supported by the Shanghai Sailing Program [21YF1416700]. The funders had no role in study design, data collection and analysis, decision to publish, or preparation of the manuscript.

**Competing interests:** The authors have declared that no competing interests exist.

As one of the intersecting research areas of Computer Vision (CV) and Natural Language Processing (NLP), research on VQA tasks has gradually increased in recent years. The main goal of visual question answering is to give computers the ability to combine visual and language modal information simultaneously for reasoning and understanding like humans. This requires the computer to perform effective feature extraction on image and text information and to effectively fuse the extracted feature information. Currently, the VQA task has also been applied to real-life scenarios, such as assisting the visually impaired, early childhood education for infants, intelligent medical care [12–14], etc. Compared with other multimodal tasks, the VQA task requires more fine-grained semantic understanding and visual reasoning on visual and text features. The model's final effect will significantly affect if the two features cannot be extracted or fused well. Therefore, establishing a closer relationship between modalities is one of the most challenging problems in VQA tasks.

The principle of the attention mechanism [15, 16] is to give different weights to different inputs by imitating the way humans observe objects so that the feature selection can be focused on and the model's prediction accuracy can be improved. The attention mechanism is a common method for establishing the relationship between modalities in multimodal tasks. Introducing the attention mechanism enables the VQA model to obtain better fine-grained image and text features, which can improve the problem of introducing noise interference information in the feature extraction of the VQA model proposed earlier and enable the model to have preliminary reasoning capabilities. Chen et al. [17] proposed to learn the visual attention of image regions from the input question to filter image key regions; Yang et al. [18] proposed a stacked attention network, which uses an iterative method to perform hierarchical attention on the image and locate it in the target image area; Lu et al. [19] proposed the concept of Co-Attention. This research uses questions to guide image attention and images to guide question attention, which is more conducive to obtaining a fine-grained image and question feature representation; Kim et al. [20] proposed Bilinear Attention Networks (BAN), which can better combine text information and visual information and learn dense relationships between problems and image regions; Yu et al. [21] proposed a Multi-modal Factorized Bilinear Pooling (MFB) method to form a more complex deep visual reasoning joint attention network through cascading; Anderson et al. [22] proposed to use Faster R-CNN [23] to extract the regional object features of the image, that is, to detect the salient objects in the picture, and then to learn the attention weight through the bottom-up-top-down attention mechanism; Research [24] explores the method of constructing the spatial position relationship of visual objects to improve the overall reasoning ability of the VQA model; The deep modular collaborative attention model MCAN proposed by Yu et al. [25], the intra-modal and inter-modal dynamic fusion model DFAF proposed by Gao et al. [26] and the dense collaborative attention model DCN proposed by Nguyen et al. [27] can effectively capture high-level information between visual and text features; The LSAT model proposed by Shen et al. [28] can effectively capture image context information while avoiding redundant information in global self-attention.

The VQA model introduced above mainly focuses on the attention mechanism, multi-modal fusion strategy, and improving model reasoning ability. However, the current attention mechanism and multi-modal fusion strategy still need to strengthen filtering noise information. In the model's training process, some irrelevant noise information will still be input, affecting the model's ability to effectively understand and reason about the input modality.

In reality, VQA models often encounter modal information that is more complex and has more noise information. For example, useless text information may still affect the model's performance in subsequent tasks after preprocessing and images with severe noise interference. Hence, it is necessary to use effective noise filtering methods to remove the interference of noise information to the system as much as possible. As shown in Fig 1(a), when the image is

**Quention: What color is the bookcase?**
**Answer: Black**

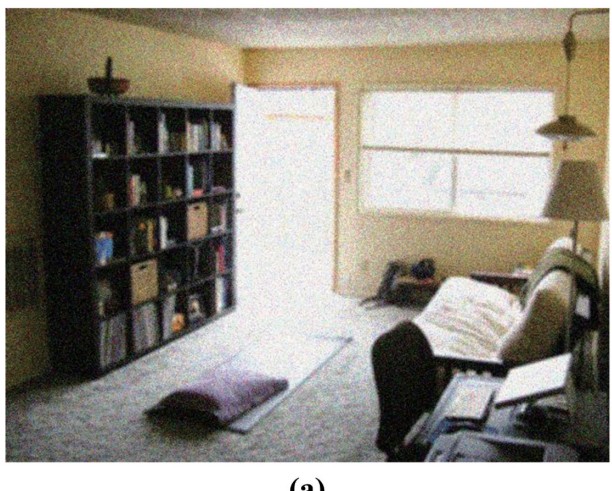

**(a)**

**Quention: How many animals in the sink?**
**Answer: 2**

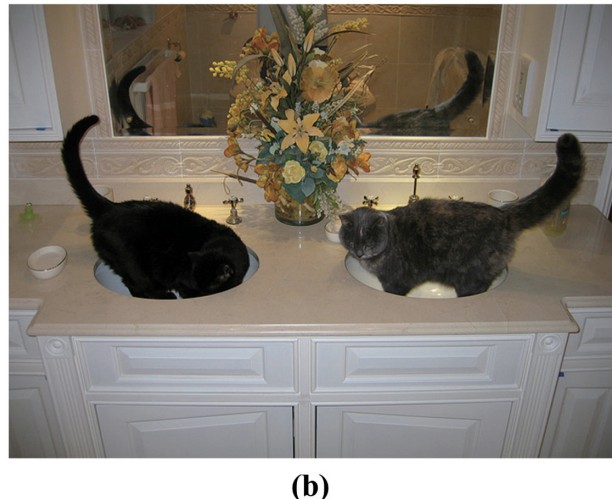

**(b)**

**Fig 1. VQA task examples.**

blurry and contains much irrelevant information, it will cause certain obstacles to the VQA model to accurately extract the information in the image, which will affect the understanding and reasoning of the model combined with text information;In Fig 1(b), the question posed to the picture is: "how many animals are there in the sink?" At this time, if the model cannot accurately extract the critical feature information "in the sink" of the text and combine it with the image for understanding and reasoning, misjudgment will quickly occur. That means only the "How many animals" in the first half of the text information is considered, so the animals in the mirror are also included in the counting range, which makes the final answer biased.

This paper considers that combining a deep collaborative attention mechanism and multi-modal fusion with the gated mechanism can further improve the overall performance of the VQA model. Based on Deep Modular Co-Attention [25] and inspired by related work [29–33], this paper proposes a novel and efficient Multi-modal Adaptive Gated Mechanism (MAGM) model for visual question answering noise information filtering problems. We introduce a gated mechanism in the encoder part, which can effectively filter the noise information in the question text and prevent it from entering the model, thereby enhancing the ability of the model to select image features combined with text features; In the decoder part, by introducing a self-guided attention gated module, the text can further filter element of the residual noise information in the image features when guiding the image features, to obtain fine-grained image features and be able to answer more complex questions. In the modal fusion module, this paper proposes an adaptive gated modal fusion mechanism, which can improve the ability of the model to adaptively control the contribution of each modal feature to the answer prediction and can also enhance the ability to maintain modality-specific semantic integrity when fusing two modality information, further enhance the interaction capability between multi-modality and improve the overall performance and accuracy of the model.

In summary, the contributions of this paper are as follows:

(1) A multi-modal adaptive gated mechanism is proposed, which can effectively filter irrelevant feature information, improve the interaction ability of different modal information, enable the model to answer complex understanding and reasoning questions better, and

improve the overall performance of the model. Obtaining more effective and fine-grained feature information as much as possible is very important for the VQA task to get the correct answer finally;

(2) The self-attention gated mechanism is introduced in the encoder part to effectively filter the text information so that the model can extract key text feature information; the self-guided attention gated network is introduced in the decoder part, which can obtain fine-grained image features so that the model can more accurately combine text and image features to understand the problem;

(3) In the modal fusion module, an adaptive gated modality fusion network is proposed, which can effectively fuse text and image modality information while improving the ability of the model to adaptively control the contribution of modal features to the predicted answer, the overall accuracy of the model is improved;

(4) The MAGM model has achieved good performance on the VQA task benchmark datasets VQA2.0 [34] and GQA [35]. The accuracy rate on the VQA2.0 dataset test-std is 71.30%, and the accuracy rate on the GQA dataset test-std is 57.57%. The best performance of the MAGM model is analyzed through ablation experiments, and the validity and interpretability of the MAGM model are further proved through visualization examples.

The remaining organizational structure of this paper is as follows: Part II introduces recent research closely related to our work briefly; Part III introduces the proposed MAGM model and describes in detail all the components of the model and the methods involved; Part IV demonstrates the effectiveness of our model through experiments and visualization examples; Part V takes a conclusion for our work and a prospect for future work.

## 2. Related works

### 2.1 Visual question answering

As a typical multimodal task of visual-language interaction, the VQA task has received more and more attention in the past few years. The VQA task aims to combine visual and text features and correctly predict the answer to a given image and related questions through understanding and reasoning. This process not only requires the model to have a deep understanding of visual information and text information at the same time but also requires the model to have a strong reasoning ability [13]. The traditional VQA model only maps the image and question features to the same high-dimensional space and performs simple fusion. This conventional method easily ignores the introduction of noise information, thus affecting the model effect. The attention mechanism is almost indispensable in most VQA-related research at present. The attention mechanism is a very effective noise-filtering method, which can improve the model's ability to understand feature information. Research [18, 36] introduced visual attention to learn image representations related to problems for the first time; While focusing on images, study [19, 37] believes that the VQA model also needs text attention in the reasoning process, so a method of using image features to guide text attention and using text to guide image attention is proposed. In addition to the attention mechanism, the multi-modal fusion strategy has a critical impact on the final effect of the VQA task [38]. The traditional multimodal fusion method splits and aligns image and problem features differently and then maps them to the public space. In recent years, related research [20, 21, 39–41] has explored more complex and effective multi-modal fusion methods.

In recent years, due to the emergence of pre-training models, the models can effectively learn more feature information and have achieved good results in various fields of deep

learning. In multimodal tasks, the intervention of pre-training methods can significantly improve the feature information extraction of each modality and the alignment and fusion between modalities. The Transformer model [42] has achieved excellent results in the NLP field, and many related studies have applied the pre-trained model based on the Transformer structure to visual language tasks, such as ViLBERT [43], VLBERT [44], LXMERT [45], Unified VLP [46], VisualBERT [47], etc.Although these pre-training models can achieve good results, their implementation requires large-scale data storage space and relatively expensive hardware resources as support. Simultaneously, the model must be migrated after the pre-training to perform the corresponding downstream tasks and evaluate the performance. At present, many related studies still adopt the end-to-end training method. Compared with the pre-training method, the end-to-end training method not only has looser restrictions on data storage space and hardware resources but can also flexibly change the model's network structure through the end-to-end training method, which can help researchers better judge the performance and performance of the model.

### 2.2 Attention mechanism

Attention mechanisms have become a common approach in multimodal tasks. Bahdanau et al. [48] first improved the attention mechanism and applied it to Neural Machine Translation (NMT) study. The attention mechanism has since been widely used in various multimodal tasks, and the performance of multimodal tasks has been greatly improved due to its intervention. In the VQA task, the attention mechanism has become essential to the VQA model. Its basic principle is to focus on specific visual areas in the image and particular words in the text to help the VQA model better answer relevant questions based on the picture.

In the VQA task, the precision and accuracy of the VQA model can be improved by using different variants of the attention mechanism to select the question text features and image features adaptively. Xu et al. [49] proposed a soft and hard attention mechanism, which became the mainstream method for VQA tasks. Subsequently, Yang et al. [18] proposed Stacked Attention Networks (SANs), which can generate multiple attention maps on images to obtain more accurate image features; Zhu et al. [50] propose to combine CNN with LSTM to generate corresponding attention maps for each word output; The DAN model proposed by Patro et al. [51] obtains one or more supporting and opposing examples by using a differential attention network to get differentiated attention regions, and its process is closer to the attention mechanism of humans in the real world; Nam et al. [37] proposed a dual attention network to obtain critical feature information by performing multi-step processing on specific regions in images and keywords in text questions; Anderson et al. [22] implemented a bottom-up attention mechanism using the object detection network Faster R-CNN [23], the principle is to divide the picture into specific single objects and screen them one by one, and then select the top-ranked custom number of suggested features in the image as visual features; Lu et al. [52] proposed a joint free-form attention mechanism, which can better expand the breadth of detection categories in object detection tasks; The MCAN model proposed by Yu et al. [25] overcomes the relationship between words in the text and the relationship between regions in the image that existed in the previous VQA model, and can better realize the interactive learning among various modalities.

### 2.3 Gate mechanism

The gate mechanism has been successfully applied in some recurrent neural network structures [53], such as LSTM, which introduces input, output, and forget gates to regulate the flow of information in and out of the module. Based on the gating mechanism, some attention

methods concentrate computing resources on the part with the wealthiest feature information [30, 54, 55].

In intra- and inter-modality learning processes and multimodal fusion process of multi-modal tasks, there may be problems that affect the interactive learning and modal fusion process due to the intervention of noise information or other irrelevant information, which will lead to a decrease in the accuracy of the output results of the model. Therefore, to effectively solve such problems, some related studies propose novel gate mechanisms in multimodal interactive learning. In the MUAN model [56], the author designed a gating model based on low-rank bilinear pooling, that is, reweighting the query matrix Q and key-value matrix K features before performing the dot product operation to select the required feature information better; John et al. [32] proposed a multimodal learning model based on a gated neural network that uses a gating mechanism similar to LSTM to fuse visual features and text features adaptively, this model is capable of adaptive selection of input parts, which helps to generate the desired output content correctly.

Although the traditional VQA model uses a gating mechanism in the process of multimodal interaction, most models ignore the problem of noise information intervention in the attention mechanism module, which quickly affects the subsequent multimodal interactive learning process and the results of the final answer prediction. Therefore, this paper adopts the self-attention-gated and self-guided-attention-gated mechanisms to deal with the problem of irrelevant information generated during the intra- and inter-modality learning process. The subsequent modal fusion and answer prediction tasks can be better completed by effectively filtering the noise information of the problem text and image features.

## 2.4 Modal fusion mechanism

For multimodal tasks, fusing multiple modal features after processing a single modal feature is crucial [57, 58]. Currently, most machine learning models still process single-modal feature data separately, perform feature alignment and other operations, and then fuse them into multi-modal results. Since different modalities have different forms of expression, and each modal looks at things from different perspectives when acquiring information, multimodal tasks have two characteristics: redundancy and complementarity. So how to reasonably handle multi-modal information fusion is the key to obtaining rich feature information.

In the related research of VQA tasks in recent years, the study [20] considered the bilinear interaction between multi-modal inputs to thoroughly learn the dense relationship between question features and image feature information; Yu et al. [39] designed a multimodal split bilinear pooling model to fuse multimodal features efficiently; Research [41] developed a multimodal tensor tucker decomposition, which can be effectively parameterized in the bilinear interaction model of text features and visual features, to perform better feature fusion.

The above research on multimodal fusion of VQA tasks needs attention to irrelevant information in high-dimensional and low-dimensional semantic information. Inspired by the study [32], we design an adaptive gated modality fusion network to perform an effective adaptive fusion of text and image two modality features. It can not only improve the ability of the model to adaptively control the contribution of the two modal features to the final prediction answer of the model but also improve the overall accuracy of the model.

## 3. Methods

In this section, the MAGM model proposed in this paper will be introduced in detail. The overall framework of the MAGM model is shown in Fig 2.

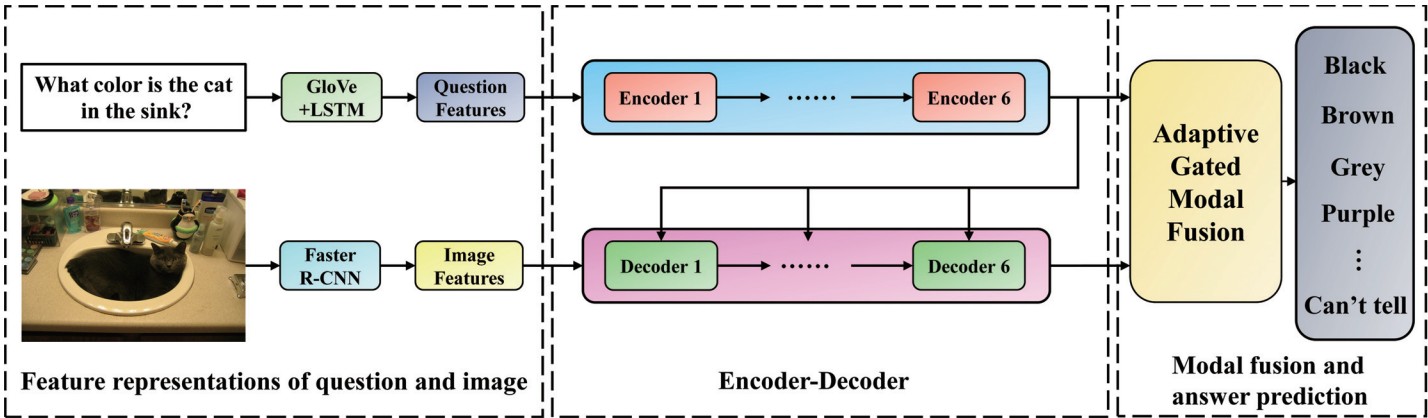

**Fig 2. The overall framework of the MAGM model proposed in this paper.**

The MAGM model comprises question and image feature extraction, Encoder-Decoder, modality fusion, and answer prediction modules. Section 3.1 first introduces how the model extracts corresponding features from input visual images and text questions. Sections 3.2 and 3.3 introduce the Encoder-Decoder structure, modality fusion, and answer prediction modules of the model, respectively.

## 3.1 Feature representation

**3.1.1 Visual feature representation.** In the image feature representation part, inspired by bottom-up attention [59], we use the pre-trained Faster R-CNN [60, 61] to extract the salient region features of the image as the visual feature representation. The output image feature representation is $Y \in R^{m \times 2048}$, where $m \in [10, 100]$ is the total number of target detection features summarized from comparative experiments in some related studies. For the convenience of comparison and consideration of hardware resources and other conditions, we set $m = 100$. In this paper, we use the linear transformation Y to make the image feature dimension consistent with the text question feature dimension, so we finally get the image feature $Y \in R^{m \times 512}$.

**3.1.2 Textual feature representation.** In the text feature representation part, consistent with the MCAN model [25], this paper uses the LSTM network to extract text question features. To make all questions have the same length, a given question is first tokenized as a sequence of words, then tokenized into words, and the number of these words is padded to a maximum value of 14, and any excess is discarded. These words are then expressed as vectors and converted into the 300-dimensional word embedding vectors by using the pre-trained 300-dimensional GloVe model [62], and the final question text features are represented as $X \in R^{n \times 512}$, where $n = 14$.

## 3.2 Encoder and decoder structure

Like the MCAN model [25], the MAGM model uses the encoder and decoder structure to model the question and image features as shown in Fig 3. The deep collaborative attention based on the Encoder-Decoder structure can simulate the intra- and inter-modality learning of problem and image features. For the VQA task, many related studies have proved that the attention model based on the Encoder-Decoder architecture is effective [24–28, 43–45]. However, whether it is intra-modal or inter-modal interactive learning, there is still uncertainty in outputting the final critical feature information. By stacking the layers of the encoder and

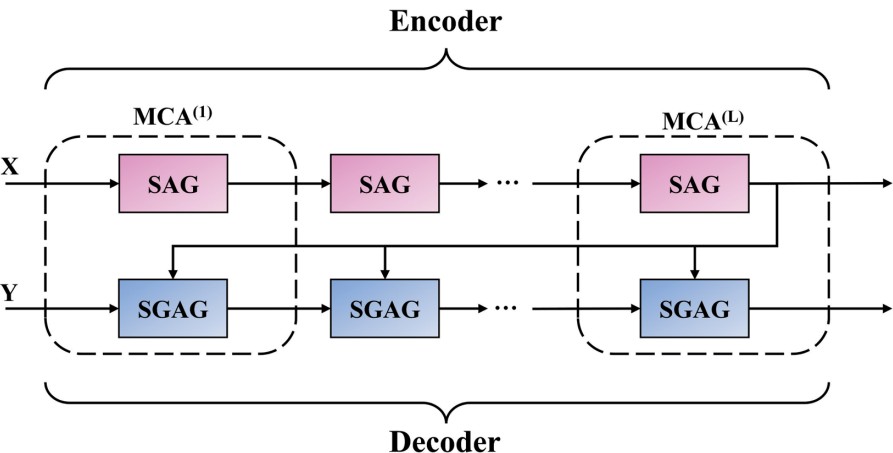

**Fig 3. The encoder and decoder structure of the MAGM model.**

decoder, finer-grained features can be obtained, but at the same time, more irrelevant noise information will be generated. Therefore, to maintain the advantages of the Encoder-Decoder structure in the MCAN model, we need to use a more effective noise filtering method to filter irrelevant redundant and invalid information while obtaining fine-grained feature information to achieve the goal of using adequate feature information more efficiently.

**3.2.1 Multi-head attention mechanism.** The Encoder and Decoder parts use a Multi-Head Attention Mechanism to enhance the representation ability of each modal feature. The mechanism first projects the query matrix Q, key-value matrix K, and value matrix V into H sub-query matrices, sub-key-value matrices, and sub-value matrices of the same dimension. Then it performs the attention operation separately in each head. Finally, the output within each head is concatenated to generate the final features. The calculation process is as follows:

$$F = MHA(Q, K, V) = \text{Concat}(head_1, \ldots\ldots, head_h)W^0 \tag{1}$$

$$head_i = Attention(QW_i^Q, KW_i^K, VW_i^V) = Softmax\left(\frac{QW_i^Q(KW_i^K)^T}{\sqrt{d}}\right)VW_i^V \tag{2}$$

Where $W^0 \in R^{d \times hd_h}$ represents the projection matrix of all heads, Concat($\bullet$) represents the operation of connecting all heads, $W_i^Q \in R^{d \times d_h}$, $W_i^K \in R^{d \times d_h}$, $W_i^V \in R^{d \times d_h}$ represent the projection matrices of the i-th head, $F \in R^{n \times d}$ represents the output feature.

**3.2.2 SAG and SGAG unit.** Fig 4(a) and 4(b) represent the SA and SGA units in the MCAN model, respectively. Inspired by study [29–31], we designed the structure of the SAG and SGAG units of the MAGM model as shown in Fig 4(c) and 4(d). Like the SA and SGA units in the MCAN model, the SAG and SGAG units can simulate the attention relationship between modalities and modalities according to different inputs. The difference is that they filter redundant feature information by adding additional gate mechanisms.

*3.2.2.1 SAG unit.* Input the question feature X to the SAG unit, learn the essential features through self-attention, and obtain more critical and rich text information features through the gate mechanism. Specifically, the SAG unit comprises three sub-layers, with X as the feature input, through the multi-head attention layer and the fully connected feed-forward network layer to simulate the intra-modal information interaction between words in the text question.

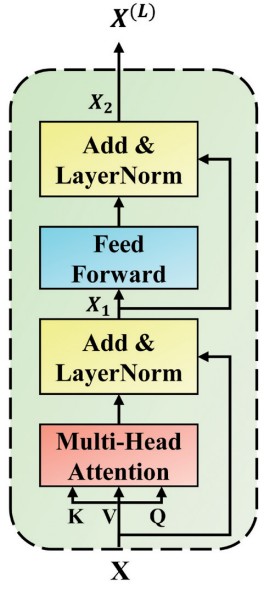

**(a) Self-Attention**

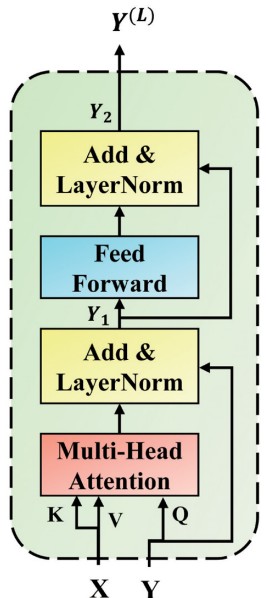

**(b) Self-Guided-Attention**

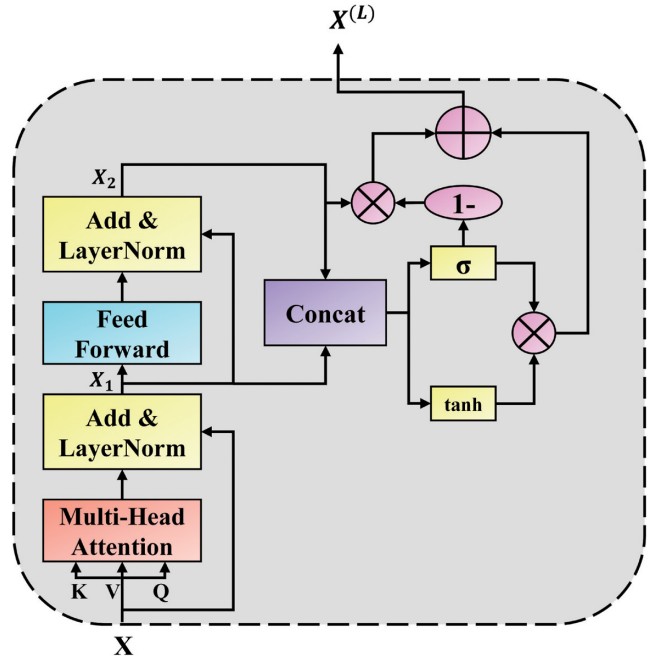

**(c) Self-Attention with Gate mechanism**

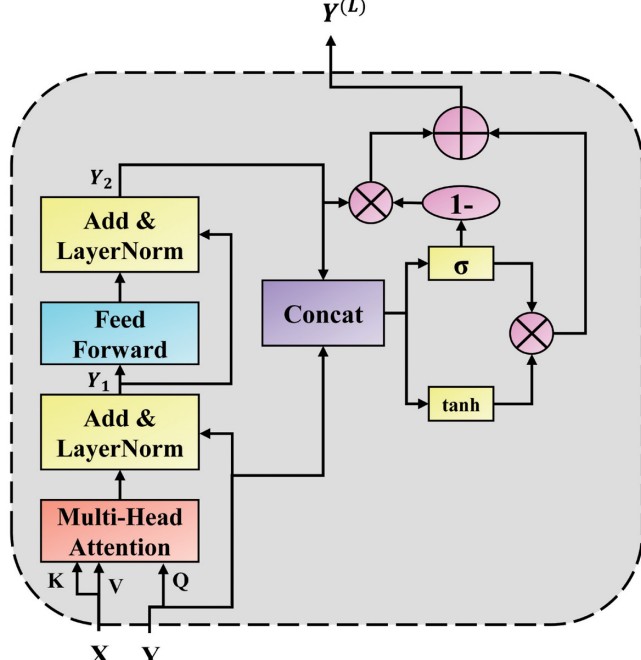

**(d) Self-Guided-Attention with Gate mechanism**

**Fig 4. Illustration of the SA and SGA unit in the MCAN model and the SAG and SGAG unit in the MAGM model.** (a) The core of the encoder in the MCAN model is the SA unit. (b) The core of the decoder in the MCAN model is the SGA unit. (c) The core of the encoder in the MAGM model is the SAG unit. (d) The core of the decoder in the MAGM model is the SGAG unit. Where $X^{(L)}$ and $Y^{(L)}$ represent question and image features of the L-th layer, respectively.

Finally, the gate mechanism is used to suppress the invalid information generated during the information interaction process within the modal.

The problem feature matrix $X_1$ participating in the output of the encoder can be expressed as:

$$X_1 = MHA(Q_x, K_x, V_x) = \text{Concat}(\text{head}_1, \cdots, \text{head}_h)W^o \tag{3}$$

$$\begin{aligned}\text{head}_i &= \text{Attention}(Q_x W_i^{Q_x}, K_x W_i^{K_x}, V_x W_i^{V_x}) \\ &= \text{Softmax}\left(\frac{Q_x W_i^{Q_x}(K_x W_i^{K_x})^T}{\sqrt{d}}\right)V_x W_i^{V_x}\end{aligned} \tag{4}$$

Where $Q_x W_i^{Q_x}, K_x W_i^{K_x}, V_x w_i^{V_x}$ represent the projection matrices. The obtained problem features $X_1$ are input to the fully connected feedforward network layer, and the question feature matrix $X_2$ can be obtained through linear transformation and activation function ReLu:

$$X_2 = FFN(X_1) = \max(0, X_1 W_1 + b_1)W_2 + b_2 \tag{5}$$

Where $W_1$, $W_2$ and $b_1$, $b_2$ represent the weight coefficient and bias variable, respectively. After obtaining the problem feature matrix $mathrmX_2$, the process of obtaining the problem feature matrix $X^{(L)}$ through the gate mechanism layer is as follows:

$$X_3 = \text{Concat}(X_2, X_1) \tag{6}$$

$$X_t = \tanh(X_3) \tag{7}$$

$$X_s = \text{sigmoid}(X_3) \tag{8}$$

$$X^{(L)} = X_t * X_s + (1 - X_s) * X_2 \tag{9}$$

Where $X^{(L)}$ represents the final output problem feature matrix after the SAG unit suppresses the invalid information generated by the self-attention mechanism of the L-th layer. Through formula (6), the output of the multi-head self-attention mechanism $X_1$ and the output of the feedforward neural network $X_2$ can be concatenated. This step can thoroughly combine the semantic dependencies captured by the multi-head self-attention mechanism and the local features of the text captured by the feedforward network; Through formula (7) and formula (8), the information of the concatenated feature vector $X_3$ can be effectively limited, so that the Gated Linear Unit (GLU) in formula (9) can be used to control the information inflow of the limited feature vector adaptively; By fusing $X_2$ in formula (9), more low- and high-level semantic features can be retained while effectively filtering irrelevant and redundant information.

Due to the short length of the question texts in the VQA task, the critical semantic features need to be fully extracted and mined. Through the above calculation process, the semantic representation ability of the feature subspace can be enhanced to better capture and represent the critical semantic information in the question texts.

*3.2.2.2 SGAG unit.* Like the SAG unit, the SGAG unit also consists of three sublayers. Taking the image feature Y and the question feature X obtained through the SAG unit as input, through the multi-head attention layer and the fully connected feedforward network layer, the question feature guides the image feature attention and simulates the intra-modal information interaction of the image area and the inter-modal information interaction between the words in the text question and the image area, and finally use the gate mechanism to suppress the

invalid redundant information generated during the intra-modal and inter-modal information interaction process. In addition, to keep the dimensionality of the question and image features consistent, we use a linear transformation. The specific formula of the SGAG unit is described as follows:

$$Y_1 = MHA(Q_y, K_x, V_x) = \ Concat \ ( \ head \ _1, \cdots, \ head \ _h)W^o \qquad (10)$$

Input the obtained image feature $Y_1$ to the fully connected feedforward network layer to obtain the image feature matrix $Y_2$:

$$Y_2 = FFN(Y_1) = max(0, Y_1 \ W_1 + b_1)W_2 + b_2 \qquad (11)$$

After obtaining the image feature matrix $Y_2$, the process of obtaining the image feature matrix $Y^{(L)}$ through the gate mechanism layer is as follows:

$$Y_3 = Concat(Y_2, Y) \qquad (12)$$

$$Y_t = tanh(Y_3) \qquad (13)$$

$$Y_s = sigmoid(Y_3) \qquad (14)$$

$$Y^{(L)} = Y_t * Y_s + (1 - Y_s) * Y_2 \qquad (15)$$

Where $Y^{(L)}$ represents the final output image feature matrix after the SGAG unit suppresses the invalid information generated by the self-guided-attention mechanism of the L-th layer. Through formula (12), the original feature input of the image Y and the output of the feedforward neural network $Y_2$ can be concatenated. This step can thoroughly combine the text information-image area interaction features captured by the multi-head self-attention mechanism and feed-forward neural network with the original image features, and then obtain rich image semantic features containing text information; Through formula (13) and formula (14), the concatenated feature vector $Y_3$ can be effectively limited, so that the GLU in formula (15) can be used to control the information inflow of the limited feature vector adaptively; By fusing $Y_2$ in formula (15), more low- and high-level semantic features can be retained while effectively filtering irrelevant and redundant information.

The above calculation process can enhance the semantic representation ability of text question-guided image features. At the same time, it can improve the semantic association between text and image and better capture the critical semantic features related to text questions in image regions.

**3.2.3 Encoder and decoder.** The Encoder of the MAGM model is composed of N-layer SAG units in a deep stacking manner to simulate the intra-modal information interaction of text question words and filter the invalid information generated during the intra-modal information interaction as much as possible through the gate mechanism. The Encoder part takes the text question feature $X = [x_1, x_2, \ldots \ldots, x_n] \in R^{n \times 512}$ as input, and after passing through the multi-head attention layer in the N-layer SAG unit, the fully connected feedforward network layer and the gate information filtering mechanism, the final question feature $X^N$ is obtained, as shown in Eq (16):

$$X^i = SAG^{[i]}(X) \qquad (16)$$

Where $i \in [1, N]$ represents the number of stacked SAG units, $X^i \in R^{n \times 512}$ represents the question feature output by the i-th layer.

The Decoder of the MAGM model is composed of N-layer SGAG units through deep stacking. While realizing the attention of image features guided by problem features, the intra-modal information interaction in the image area and the inter-modal information interaction between the words in the text question and the image area are simulated. Finally, as much as possible, the gate mechanism filters the invalid redundant information generated in the intra- and inter-modality learning process. The Decoder part takes the image feature $Y = [y_1, y_2, \ldots\ldots, y_n] \in R^{n \times 512}$ and the question feature obtained from the Encoder part as input. After passing through the multi-head attention layer in the N-layer SGAG unit, the fully connected feedforward network layer, and the gate information filtering mechanism, the final image feature $Y^N$ is obtained, as shown in Eq (17):

$$Y^j = SGAG^{[j]}(Y, X^i) \tag{17}$$

Where $j \in [1, N]$ represents the number of stacked SGAG units, $Y^j \in R^{n \times 512}$ represents the image feature output by the j-th layer.

### 3.3 Modal feature fusion and answer prediction

The question feature X and the image feature Y containing the most significant feature value output by the encoder and decoder structure, provide rich and fine-grained weight information for text words and image regions, respectively. At this time, we first design a two-layer MLP (FC(512)-Relu-Dropout(0.1)-FC(1)) to fuse the problem feature X and the image feature Y. Specifically, we input the image feature Y into the MLP to calculate the attention weight value through the softmax function, and then multiply and sum the image features of each region to obtain the fused image feature, the equations are as follows:

$$\lambda = softmax(MLP(Y^{(L)})) \tag{18}$$

$$\overline{Y} = \sum_{j=1}^{n} \lambda_i y_j^{(L)} \tag{19}$$

Where $\lambda = [\lambda_1, \cdots \lambda_n] \in R^n$ represents the image learning weight. Similarly, we can get the fused question feature $\overline{X}$.

After the above steps, we get the preliminarily fused question feature $\overline{X}$ and image feature $\overline{Y}$. However, there may still be some noise in the feature information, so it needs to filter further to make more valuable features for the model to make final predictions. Inspired by research [32], we designed the adaptive gated feature fusion structure shown in Fig 5 to reduce the relevant redundant information in the multimodal fusion mechanism, and its calculation process is as follows:

$$h_q = sigmoid(\overline{X}) \tag{20}$$

$$h_i = sigmoid(\overline{Y}) \tag{21}$$

$$h = (h_t * h_t) + (h_i * h_i) \tag{22}$$

$$z = sigmoid(h) \tag{23}$$

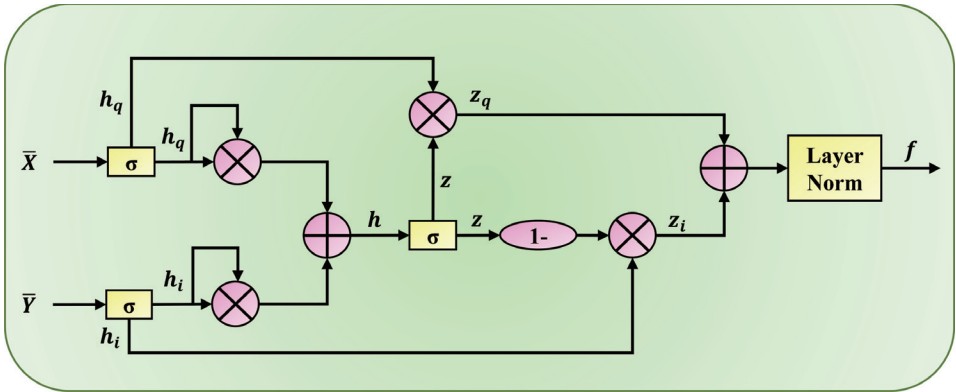

**Fig 5. The adaptive gated feature fusion structure of the MAGM model.**

$$z_q = z * h_q \tag{24}$$

$$z_i = (1 - z) * h_i \tag{25}$$

$$f = \text{LayerNorm}(W_q^T z_q + W_i^T z_i) \tag{26}$$

Where f represents the fusion feature of text and image. $W_q^T$, $W_i^T$ are linear projection matrices. Through formula (20), formula (21), and formula (23), deep-level and fine-grained information filtering is carried out on the preliminarily fused problem feature $\overline{X}$, image feature $\overline{Y}$, and their weighted sum feature h. Finally, through formula (24–26), the adaptive fusion of text and image feature representations can be realized so that the model can thoroughly learn the high-level modality fusion semantic feature representations.

After the above calculation steps, f is passed to the linear activation function Linear, and the sigmoid function is used to classify the answer [63]. During training, we use the Binary Cross-Entropy (BCE) function as the loss function [59]:

$$s = \text{sigmoid}(\text{Linear}(f)) \tag{27}$$

Where s represents the score weight of candidate answers, and the candidate answer with the largest weight is selected as the prediction result. Finally, use the BCE function as the loss function to train N answer categories:

$$N = \sum_{i=1}^{n} \gamma_i \log(s_i) + (1 - \gamma_i) \log(1 - s_i) \tag{28}$$

Where N is the size of the candidate answer set, $s_i$ is the prediction score of the model for each candidate answer, and $\gamma_i$ is the soft score that provides the answer in the dataset.

## 4. Experiments

This section is divided into four parts to prove the validity and rationality of the MAGM proposed in this paper. Section 4.1 introduces the benchmark datasets VQA2.0 [34] and GQA [35] used in the experiment; In Section 4.2, the specific details of the experiment and

hyperparameter settings are introduced; Section 4.3 discusses the ablation experiment performed on the MAGM model to explore the influence of the core part of the model on the overall model; In Section 4.4, the performance results of the MAGM model on the VQA2.0 dataset and the GQA dataset are compared with the state-of-the-art models; In Section 4.5, the performance advantage of our proposed MAGM model is demonstrated by visually comparing the attention with different models. All experiments in this paper are based on Ubuntu 20.04 system, GPU is RTX3090, the CUDA version is 11.7, and the deep learning framework is PyTorch.

## 4.1 Datasets

**VQA2.0** [34]: This dataset is currently the most commonly used benchmark dataset for VQA tasks, consisting of images and associated Question-Answer (QA) pairs from the MS-COCO dataset [64]. Each picture corresponds to at least three questions, corresponding to 10 answers. The VQA2.0 dataset consists of three parts: training set, verification set, and testing set, and the data distribution is as follows:

- Training set: Contains 82783 images and 443757 QA pairs for training.

- Validation set: Contains 40504 images and 214354 QA pairs for validation.

- Test set: Contains 81434 images and 447793 QA pairs for testing.

Additionally, the test set is split into two subsets: test-dev and test-standard, according to the ratio of 1:3. The questions in the VQA2.0 dataset can be divided into three types: "Yes/No" (Y/N), "Count" (Number), and "Other" according to the different categories of answers.

**GQA** [35]: The dataset consists of 113K images and 22M questions generated from these images. Compared with the VQA 2.0 dataset, the GQA dataset contains more questions that require multi-step reasoning, and its answer distribution is more balanced. About 94% of the questions require multi-step rationale, and 51% need to query the relationship between objects. In addition to the standard accuracy metrics, the authors of this dataset added some new metrics, such as consistency, credibility, validity, and distribution distance, to better help evaluate model performance.

This paper presents the experimental results on the test-dev and test-standard subsets of the benchmark dataset VQA2.0 and GQA, respectively.

## 4.2 Implementation details

In our experiments, the dimension size of the hidden layer in Scaled Dot-Product Attention is set to d = 512, the number of attention heads H of Multi-head Attention is 8, and the feature dimension of each head output is d/H = 64. Referring to the suggestion of research [40], set the structure of the feed-forward layer as FC(4d)—ReLU—Dropout(0.1) – FC(d)). The structure of the multi-layer perceptron used to calculate the features of interest as FC(d)—ReLU—Dropout(0.1) – FC(1)), Where ReLU is the activation function, and the dropout rate is used to prevent overfitting during the training process. We use the AdamW optimizer [65] ($\beta_1 = 0.9$, $\beta_2 = 0.98$) to train the MAGM model. According to the default parameters, the epoch number is set to 13 (epoch is set to 11 on the GQA dataset), the batch size is 64, and the binary cross entropy function BCE is used as the loss function. The learning rate is set to $min(2.5te^{-5}, 1e^{-4})$, where t represents the current epoch number starting from 1, after 10 epoch training, the learning rate drops to 1/5 of the current learning rate every two epochs.

The VQA2.0 training set used in this experiment includes the train subset, the val subset, and the additional data vg subset. The training split method is "train+val+vg", where the vg

subset is the QA pair from the Visual Genome dataset [66]; The GQA training set includes the train subset and the val subset, the training split method is "train+val".

## 4.3 Ablation studies

This section mainly discusses choosing the optimal parameters and proving the validity and interpretability of the model. In this paper, different MAGM variant models are designed, training on the VQA 2.0 dataset using the "train+val+vg" split method and training with the "train+val" split method on the GQA dataset, and testing on the test-dev test subset of the two datasets to get the results. In Section 4.3.1, we explore the effects of various MAGM variant models using SAG units, SGAG units, and multimodal adaptive gated feature fusion structures under a fixed number of Encoder-Decoder layers; In Section 4.3.2, on the premise of using the multimodal adaptive gated feature fusion mechanism, different Encoder-Decoder layers are set to explore the best performance of the MAGM model.

**4.3.1 MAGM variants.** The experimental results of the performance comparison between different MAGM model variants designed in this paper on the VQA2.0 and GQA datasets are shown in Tables 1 and 2, respectively. The experimental results in Table 1 are training with the split method of "train+val+vg" and obtained through online testing on the test-dev subset of the VQA2.0 dataset; The experimental results in Table 2 are training with the split method of "train+val" and obtained through online testing on the test-dev subset of the GQA dataset.

The MAGM model uses three strategies to filter irrelevant and redundant information: a gate mechanism to the SA unit and SGA unit in the original MCAN model and a modal adaptive gated fusion mechanism to the modal feature fusion part, denoted by SAG, SGAG, and MFG respectively. "MAGM-SAG" in Tables 1 and 2 indicates that the MAGM model only adds a gating mechanism to the SA unit of the original MCAN model, the original SGA unit, and the modal feature fusion part keeps the original MCAN model structure unchanged, and so on for the rest. For the ablation experiment of the MAGM model on the GQA dataset in Table 2, we show in Fig 6 that the "MAGM-SAG+SGAG+MFG" model is trained on the GQA dataset using the training split method of "train+val", And record the variation of the training accuracy with the number of training rounds when the accuracy of each epoch is verified on the local validation set.

Fig 6 shows that when the model is trained to the 9th round, it can achieve the best effect on the "Accuracy" index. As the number of training rounds increases, the accuracy gradually decreases and tends to balance. Therefore, all the MAGM variant models on the GQA dataset in Table 2 and the MAGM models on the GQA dataset for comparison in Table 5 use the models trained in epoch 9. The experimental results in Tables 1 and 2 show that when the MAGM model uses the SAG, SGAG, and MFG modules separately, the overall invalid information

**Table 1. The performance of different variant models of MAGM on the VQA2.0 dataset.**

| Model | Y/N | Number | Other | All |
|---|---|---|---|---|
| MCAN [25] | 86.82 | 53.26 | 60.72 | 70.63 |
| MAGM-SAG | 86.77 | 53.32 | 60.97 | 70.73 |
| MAGM-SGAG | 86.84 | 53.39 | 60.99 | 70.78 |
| MAGM-MFG | **87.05** | 53.01 | 61 | 70.82 |
| MAGM-SAG+SGAG | 87.02 | 53.35 | 60.87 | 70.79 |
| MAGM-SAG+MFG | 87.02 | **53.6** | 60.9 | 70.83 |
| MAGM-SGAG+MFG | 87.03 | 53.36 | 61.08 | 70.89 |
| **MAGM-SAG+SGAG+MFG** | **87.05** | 53.38 | **61.18** | **70.95** |

**Table 2. The performance of different variant models of MAGM on the GQA dataset.**

| Model | Accuracy | Open | Binary | Validity | Plausibility | Consistency |
|---|---|---|---|---|---|---|
| LCGN [67] | 55.8 | - | - | - | - | - |
| MAGM-SAG | 56.72 | 40.13 | 75.59 | 96.7 | 85.21 | 86.63 |
| MAGM-SGAG | 56.36 | 40.88 | 73.96 | 96.95 | 85.37 | 84.75 |
| MAGM-MFG | 57.18 | 40.93 | 75.67 | 96.71 | 85.51 | 87.92 |
| MAGM-SAG+SGAG | 57.44 | 41.29 | 75.82 | 96.9 | **85.53** | 87.55 |
| MAGM-SAG+MFG | 57.66 | 41.22 | **76.36** | **97.02** | 85.44 | 87.28 |
| MAGM-SGAG+MFG | 56.77 | 40.81 | 74.92 | 96.81 | 85.31 | 87.53 |
| **MAGM-SAG+SGAG+MFG** | **57.73** | **41.97** | 75.67 | 96.98 | 85.43 | **88.34** |

filtering ability of the model has been improved to a certain extent. When the MFG module is used alone, the model's overall performance is greatly enhanced. When the MFG module is used with the SAG and SGAG modules, it can further reduce the redundant information generated by the text and image features in the modal fusion process.

On the VQA2.0 dataset, although the MAGM model may lack the ability to perform practical in-depth reasoning on the combination of text and image features after using the three modules of SAG, SGAG, and MFG at the same time, the model's performance on the "Number" category problems is lacking. However, the MAGM model can achieve the best effect on the "Other" type of problems, which proves that the MAGM model can better understand and obtain the text feature information in the "Other" category of problems after effectively filtering the redundant noise information. At the same time, it can better make the question features effectively guide the image features and achieve better results in the "Other" category

The Accuracy changes during "MAGM-SAG+SGAG+MFG" training on GQA

**Fig 6. The accuracy changes during the "MAGM-SAG+SGAG+MFG" model training on the GQA dataset.**

**Table 3. The MAGM model performance with different layers of encoder and decoder.**

| L | Y/N | Number | Other | All |
|---|---|---|---|---|
| 2 | 86.1 | 52.22 | 60.7 | 70.2 |
| 4 | 86.62 | 52.88 | 60.87 | 70.57 |
| 5 | 86.93 | 53.35 | 61.11 | 70.86 |
| **6** | 87.05 | **53.38** | **61.18** | **70.95** |
| 7 | **87.12** | 53.35 | 60.99 | 70.89 |
| 8 | 87.06 | 53.3 | 60.9 | 70.81 |

questions; Similarly, on the GQA dataset, when the MAGM model uses three modules at the same time, some indicators may not be optimal due to insufficient in-depth reasoning and understanding capabilities. However, the "Accuracy", "Open", and "Consistency" indicators have all achieved sound improvement effects, which further proves that the MAGM model is effective in filtering irrelevant and redundant information. "MAGM-SAG+SGAG+MFG" is the final model used in this paper.

Through the comparison of the experimental ablation results in this section, the MAGM model proposed in this paper can not only effectively filter the noise information of text features and image features, but also adaptively filter irrelevant and redundant information in the process of modality fusion, making the model exhibit excellent performance on both the VQA2.0 dataset and the GQA dataset.

**4.3.2 MAGM parameter ablation.** This section mainly discusses the selection of parameters used in the MAGM model. Since the MAGM model uses an Encoder-Decoder architecture, which includes SAG and SGAG modules, different Encoder-Decoder layers are used to verify the model performance using the MFG module. The experimental results of this part are all trained in the split method of "train+val+vg" and obtained through online testing on the test-dev subset of the VQA2.0 dataset. The experimental results are shown in Table 3.

From the results in Table 3, when the number of layers L = 6, the MAGM model can achieve the best results in the Number and Other problems and the overall performance. When the number of layers continues to increase, the overall performance of the model begins to decline, so the MAGM model is set to L = 6 in the experiment results in Tables 1, 2, 4 and 5.

## 4.4 Performance comparison

**4.4.1 The results of performance comparison with the SOTA methods on VQA 2.0.** This paper compares the performance of the MAGM model with the current state-of-the-art models on the VQA 2.0 dataset, and the results are shown in Table 4.

As shown in Table 4, we compare the MAGM model with the current SOTA model, and the last row of Table 4 is the test result of the MAGM model proposed in this paper. The bilinear attention network BAN [20] considers the bilinear interaction between multimodal inputs to utilize the question feature and image feature information fully; BAN-Counter [20] combines BAN with Counter [20], which is a neural network component, can further improve the accuracy of the model on Number-type problems through the robust counting function; Bottom-up [59] and Bottom-up+MFH [25] combine regional visual features with question-guided visual attention; LSAT-R [28] model considers local self-attention, which can effectively avoid redundant information in global self-attention ("-R" indicates that the LSAT model is trained on the VQA2.0 dataset using the same region image features as the MAGM model and other SOTA models for comparison); Unified VLP [46] is a bidirectional and seq2seq-based unified visual-language pre-training model that can be fine-tuned for visual-language generation and understanding tasks. The pre-

**Table 4. The results of MAGM model performance comparison with the SOTA methods on the VQA2.0 dataset.**

| Methods | Test-dev | | | | Test-std |
|---|---|---|---|---|---|
| | Y/N | Num | Other | All | All |
| Language only [34] | - | - | - | - | 44.26 |
| LSTM+CNN [34] | - | - | - | - | 54.22 |
| MCB reported in [34] | - | - | - | - | 62.27 |
| Bottom-up [59] | 81.82 | 44.21 | 56.05 | 65.32 | 65.67 |
| MFH [25] | 85.31 | 49.56 | 59.89 | 68.76 | - |
| Bottom-up+MFH [25] | 84.27 | 49.56 | 59.89 | 68.76 | - |
| DCN [27] | 83.50 | 46.60 | 56.72 | 66.60 | 67.00 |
| BAN [20] | 85.42 | 50.93 | 60.26 | 69.52 | - |
| BAN-Counter [20] | 85.42 | 54.04 | 60.52 | 70.04 | 70.35 |
| VRR [68] | 83.31 | 45.51 | 58.41 | 67.20 | 67.34 |
| DFAF [26] | 86.09 | 53.32 | 60.49 | 70.22 | 70.34 |
| MuRel [25] | 84.77 | 49.84 | 57.85 | 68.03 | 68.41 |
| ReGAT [69] | 86.08 | **54.42** | 60.33 | 70.27 | 70.58 |
| MCAN [25] | 86.82 | 53.26 | 60.72 | 70.63 | 70.90 |
| MUAN [56] | 86.77 | 54.40 | 60.89 | 70.82 | 71.10 |
| LSAT-R [28] | 87.06 | 53.32 | 61.04 | 70.88 | 71.13 |
| Unified VLP [46] | **87.20** | 52.10 | 60.30 | 70.50 | 70.70 |
| ViLBERT [43] | - | - | - | 70.55 | 70.92 |
| VisualBERT [47] | - | - | - | 70.80 | 71.00 |
| **MAGM (ours)** | 87.05 | 53.38 | **61.18** | **70.95** | **71.30** |

training models ViLBERT [43] and VisualBERT [47] use the BERT architecture, where VisualBERT is a single-stream model, and ViLBERT is a two-stream model.

According to the results in Table 4, the overall performance of the MAGM model proposed in this paper is better than that of previous models focusing on visual relational reasoning or modality fusion. Most of its indicators are better than existing advanced methods. Compared with the MCAN model, the accuracy of the MAGM model in the "Yes/No" category increased by 0.23%, in the "Number" category by 0.12%, in the "Other" category by 0.46%, and the overall accuracy in "All" improved by 0.32% and 0.40% on test-dev and test-std, respectively. Especially on "Other" category problems, the accuracy rate exceeds 61%, demonstrating that the attention gate mechanism and multi-modal adaptive gated fusion strategy proposed in this paper effectively filter noise information.

**4.4.2 The results of performance comparison with the SOTA methods on GQA.** The results of the comparison between the MAGM model and the existing state-of-the-art models on the test-standard subset of the GQA dataset are shown in Table 5.

As shown in Table 5, we compared the MAGM model with the current SOTA model. "Human" represents the performance of humans on the GQA dataset, which can be considered as the upper bound of the performance of the VQA model on the dataset. The last row of Table 5 is the test results of the MAGM model proposed in this paper. MAC [35] is a milestone model on the CLEVR dataset [73], which decomposes a task into a series of sequential reasoning subtasks; SceneGCN [70] uses a graph neural network to model visual object regions, which jointly infers visible object semantic relations and attribute relations for VQA tasks; GRNs [71] and LCGN [67] use the message passing strategy to perform relational inference for VQA tasks; GMA [72] focuses on constructing graphs from question and image regions, and seeks a structured alignment of the two modalities.

**Table 5. The results of MAGM model performance comparison with the SOTA methods on the GQA dataset.**

| Methods | Accuracy | Open | Binary | Validity | Plausibility | Consistency |
|---|---|---|---|---|---|---|
| Human [35] | 89.30 | 87.40 | 91.20 | 98.90 | 97.20 | 98.40 |
| LSTM+CNN [35] | 46.55 | 31.80 | 63.26 | 96.02 | 84.25 | 74.57 |
| Bottom-up [35] | 49.74 | 34.83 | 66.64 | 96.18 | 84.57 | 78.71 |
| MAC [35] | 54.06 | 38.91 | 71.23 | 96.16 | 84.48 | 81.59 |
| SceneGCN [70] | 54.56 | 40.63 | 70.33 | 95.90 | 84.23 | 83.49 |
| BAN [20] | 56.19 | 41.13 | 73.31 | **96.77** | **85.58** | 84.64 |
| LCGN [67] | 56.10 | - | - | - | - | - |
| GRNs [71] | 57.04 | 41.24 | 74.93 | 96.14 | 84.68 | 87.41 |
| GMA [72] | 57.26 | **42.30** | 73.20 | 96.41 | 85.05 | 83.95 |
| **MAGM (ours)** | **57.57** | 41.89 | **75.33** | 96.17 | 84.70 | **88.05** |

According to the results in Table 5, on the GQA dataset, the MAGM model has better accuracy than the current SOTA model. Compared with GMA [72], the "Accuracy", "Binary", and "Consistency" indexes on the MAGM model have increased by 0.31%, 2.13%, and 4.10%, respectively. It is proved that the attention gate mechanism and the multimodal adaptive gated fusion strategy proposed in this paper effectively improve the VQA model's reasoning ability.

## 4.5 Visualization

Fig 7(a) shows the visualization results in the VQA2.0 dataset. It can be seen from the first picture that the MAGM model can better capture the critical feature information in the text and image, thus correctly judging that the color of the tennis racket in the athlete's hand is red and white, while the MCAN model cannot answer correctly; It can be seen from the second figure that after filtering irrelevant noise information, the MAGM model can better pay attention to the image region features according to the text problem. Fig 7(b) shows an example of visualization in the GQA dataset. Most of the questions in the GQA dataset require the model to have a thorough understanding and reasoning ability. For instance, in the first picture, the model needs to understand the spatial concept of "in front of the table" in combination with text questions and then needs to reason about the spatial position relationship centered on the table in the picture, and finally determine the visual object "TV" as the correct answer; In the second picture, there is only a tiny part of the monkey's head in the view, which requires the model to have a specific image feature capture ability to recognize it as a monkey and answer the question correctly; In the last picture, like the comparison model, the answer of the MAGM model is also wrong, because this question requires a combination of reasoning and external knowledge understanding to answer. Therefore, even if the model extracts enough text and image features, it still does not have a deep understanding of the problem nor a sufficient account of external knowledge, so it is difficult to answer such questions correctly.

## 5. Conclusion

This paper studies the effectiveness of introducing a gating mechanism in intra- and inter-modal interactive learning and adding a multi-modal adaptive gated mechanism in the process of multi-modal fusion to improve the overall performance of the VQA task. Compared with the traditional attention unit, the self-attention gated unit, and self-guided attention gated unit proposed in this paper can effectively improve the acquisition of text question features and image features information, enhance the guidance of text features to image features, and reduce the introduction of irrelevant information. Finally, this paper adds an adaptive gating

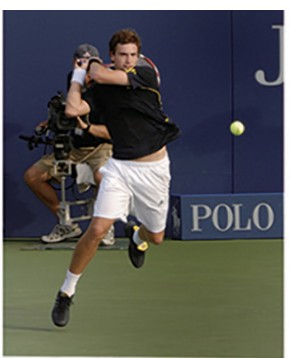 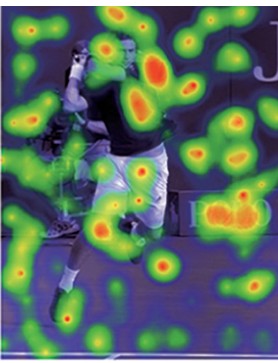 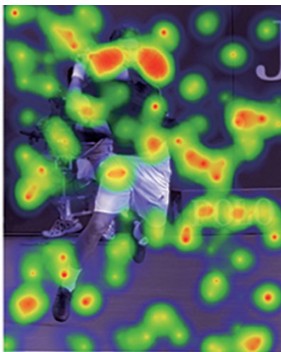

**Q: What color is the tennis racket in the hands of athletes?**
**Ground Truth: Red and white**

MCAN: Red ✗

MGMN: Red and white ✓

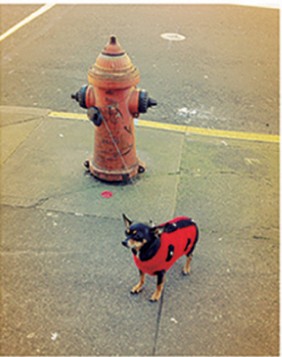 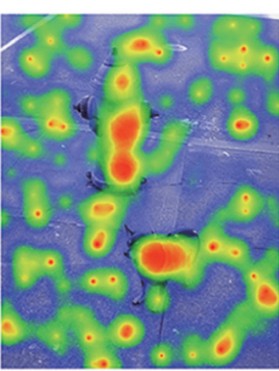 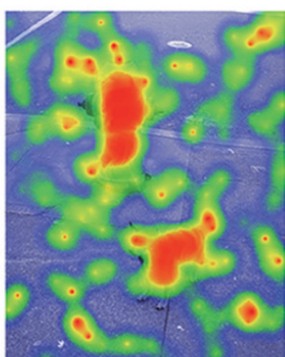

**Q: Is there a fire hydrant behind the dog?**
**Ground Truth: Yes**

MCAN: Yes ✓

MGMN: Yes ✓

(a)

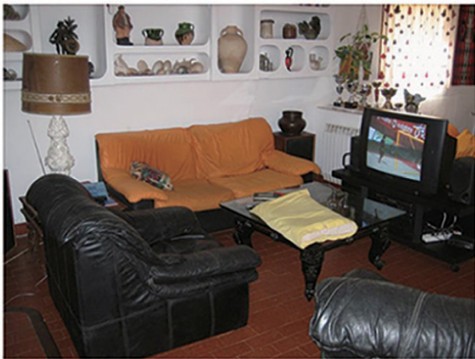 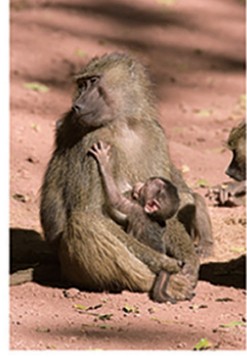 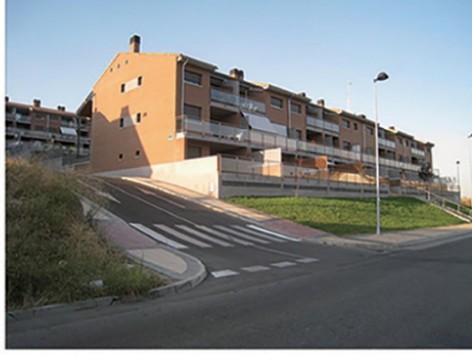

**Q: What is in front of the table on the floor?**
**Ground Truth: TV**
**BAN: TV ✓**
**MGMN: TV ✓**

**Q: How many monkeys are there in the picture?**
**Ground Truth: 3**
**BAN: 2 ✗**
**MGMN: 3 ✓**

**Q: What is the purpose of the road in the picture?**
**Ground Truth: For the passage of vehicles**
**BAN: Unknown ✗**
**MGMN: Unknown ✗**

(b)

**Fig 7. The results of model visualization on VQA2.0 and GQA datasets.**

modal feature fusion structure in the modal fusion process, which can further improve the overall irrelevant noise information filtering ability of the model, better retain fine-grained modal feature information, and overall improve the ability of the model to adaptively control the contribution of each modal feature to the predicted answer. The relevant experiments on the MAGM model proposed in this paper on the VQA task benchmark dataset VQA2.0 and GQA prove the interpretability and effectiveness of the MAGM model. It is helpful to explore further the irrelevant noise information filtering problem of the VQA task model in future work research.

The MAGM model mainly focuses on obtaining finer-grained modal features and filtering more irrelevant noise information during intra- and inter-modal interactive learning and modal fusion. Currently, the difficulty of the VQA task mainly lies in how to effectively model the spatial position relationship of visual objects and other interactive relationships (such as behavioral relationships representing action interactions, etc.) to improve the overall reasoning ability of the VQA model. Therefore, in future work, we will focus on exploring the interactive relationship between visual objects in the VQA task and further explore the application and combination of the irrelevant noise information filtering method proposed in this paper with the interactive relationship between the visual objects.

## Author Contributions

**Conceptualization:** Lin Zhang.

**Funding acquisition:** Lin Zhang.

**Methodology:** Yangshuyi Xu.

**Resources:** Xiang Shen.

**Validation:** Yangshuyi Xu, Xiang Shen.

**Writing – original draft:** Yangshuyi Xu.

**Writing – review & editing:** Yangshuyi Xu, Lin Zhang, Xiang Shen.

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
