## [Decision Letter · Decision Letter 0]

10 Apr 2023

PONE-D-23-05435Multi-modal Adaptive Gated Mechanism for Visual Question AnsweringPLOS ONE

Dear Dr. Zhang,

Thank you for submitting your manuscript to PLOS ONE. After careful consideration, we feel that it has merit but does not fully meet PLOS ONE’s publication criteria as it currently stands. Therefore, we invite you to submit a revised version of the manuscript that addresses the points raised during the review process.

Please address the comments of the reviewers and resubmit the paper.==============================

We look forward to receiving your revised manuscript.

Kind regards,

Sriparna Saha, PhD

Academic Editor

PLOS ONE

Journal Requirements:

5. We note that Figure 7 includes an image of a [patient / participant / in the study]. 

6. We note that Figures 1 and 7 in your submission contain copyrighted images. All PLOS content is published under the Creative Commons Attribution License (CC BY 4.0), which means that the manuscript, images, and Supporting Information files will be freely available online, and any third party is permitted to access, download, copy, distribute, and use these materials in any way, even commercially, with proper attribution. For more information, see our copyright guidelines: http://journals.plos.org/plosone/s/licenses-and-copyright.

a. You may seek permission from the original copyright holder of Figures 1 and 7  to publish the content specifically under the CC BY 4.0 license. 

Reviewers' comments:

Reviewer's Responses to Questions

**Comments to the Author**

1. Is the manuscript technically sound, and do the data support the conclusions?

Reviewer #1: Partly

Reviewer #2: Yes

2. Has the statistical analysis been performed appropriately and rigorously? 

Reviewer #1: Yes

Reviewer #2: Yes

3. Have the authors made all data underlying the findings in their manuscript fully available?

Reviewer #1: Yes

Reviewer #2: Yes

4. Is the manuscript presented in an intelligible fashion and written in standard English?

Reviewer #1: Yes

Reviewer #2: Yes

5. Review Comments to the Author

Reviewer #1: Please can you explain in more detail how the gating mechanism is used to reduce noise.

In equation 28, the left hand side of equation should be BCELoss not "N"

In equation 28, what does "n" represent. It should be replaced with N (as N represent number of answer categories.)

Reviewer #2: This article improves the performance of the VQA task by an introduction of a gating mechanism in intra- and inter-modal interactive learning, along with a multi-modal adaptive gated mechanism during multi-modal fusion. To better control the contribution of each modal feature to the predicted answer, the authors propose utilizing the self-attention gated unit and the self-guided attention gated unit. Experiment results on the VQA2.0 and GQA benchmark datasets show the relevence of the proposed model. In general, the contribution of this work is significant. I recommend this work to be published.

6. PLOS authors have the option to publish the peer review history of their article (what does this mean?). If published, this will include your full peer review and any attached files.

Reviewer #1: No

Reviewer #2: **Yes: **Thien Nguyen

---

## [Author Response · Author response to Decision Letter 0]

14 Apr 2023

Dear Editor,

Thank you for your letter and the reviewers’ comments concerning our manuscript, “Multi-modal Adaptive Gated Mechanism for Visual Question Answering” (PONE-D-23-05435). These comments are all valuable and helpful for revising and improving our manuscript and the important guiding significance to our research. We have carefully studied the comments point-by-point and revised the paper accordingly.

Thanks again for your patience and guidance!

Best regards,

Yangshuyi Xu, Lin Zhang, Xiang Shen

Reviewer's Responses to Questions

Comments to the Author

1. Is the manuscript technically sound, and do the data support the conclusions?

Reviewer #1: Partly

Reviewer #2: Yes

Response: Thank you for your comments and suggestions on our paper. We have carefully considered and responded to all the questions we asked you below and marked them in blue font in the manuscript for easy review.

2. Has the statistical analysis been performed appropriately and rigorously?

Reviewer #1: Yes

Reviewer #2: Yes

Response: We are very grateful for your affirmation of our work.

3. Have the authors made all data underlying the findings in their manuscript fully available?

Reviewer #1: Yes

Reviewer #2: Yes

Response: We are very grateful for your affirmation of our work.

4. Is the manuscript presented in an intelligible fashion and written in standard English?

Reviewer #1: Yes

Reviewer #2: Yes

Response: We are very grateful for your affirmation of our work.

5. Review Comments to the Author

Reviewer #1: Please can you explain in more detail how the gating mechanism is used to reduce noise.

Response: First, thank you for reviewing our paper and giving suggestions. We have carefully considered your questions and added new content in lines 341-353, 380-391, and 450-454 of the manuscript, marked in blue font, and we use yellow background and strikethrough to mark older content, so you can better distinguish them.

In subsection 3.2.2.1, we introduce a gating mechanism in the SAG unit. Through formula (6), the output of the multi-head self-attention mechanism〖 X〗_(1 )and the output of the feedforward neural network X_(2 )can be concatenated. This step can thoroughly combine the semantic dependencies captured by the multi-head self-attention mechanism and the local features of the text captured by the feedforward network; Through formula (7) and formula (8), the information of the concatenated feature vector X_3 can be effectively limited, so that the Gated Linear Unit (GLU) in formula (9) can be used to control the information inflow of the limited feature vector adaptively; By fusing X_2 in formula (9), more low- and high-level semantic features can be retained while effectively filtering irrelevant and redundant information. Due to the short length of the question texts in the VQA task, the critical semantic features need to be fully extracted and mined. Through the above calculation process, the semantic representation ability of the feature subspace can be enhanced to better capture and represent the critical semantic information in the question texts. We marked the relevant revisions in a blue font at lines 352-365 of the manuscript.

In subsection 3.2.2.2, we introduce a gating mechanism in the SGAG unit. Through formula (12), the original feature input of the image Y and the output of the feedforward neural network Y_2 can be concatenated. This step can thoroughly combine the text information-image area interaction features captured by the multi-head self-attention mechanism and feed-forward neural network with the original image features, and then obtain rich image semantic features containing text information; Through formula (13) and formula (14), the concatenated feature vector Y_3 can be effectively limited, so that the GLU in formula (15) can be used to control the information inflow of the limited feature vector adaptively; By fusing Y_2 in formula (15), more low- and high-level semantic features can be retained while effectively filtering irrelevant and redundant information. The above calculation process can enhance the semantic representation ability of text question-guided image features. At the same time, it can improve the semantic association between text and image and better capture the critical semantic features related to text questions in image regions. We marked the relevant revisions in a blue font at lines 387-400 of the manuscript.

In Section 3.3, we introduce an adaptive gated feature fusion structure. Through formula (20), formula (21), and formula (23), deep-level and fine-grained information filtering is carried out on the preliminarily fused problem feature X ®, image feature Y ®, and their weighted sum feature h. Finally, through formula (24-26), the adaptive fusion of text and image feature representations can be realized so that the model can thoroughly learn the high-level modality fusion semantic feature representations. We marked the relevant revisions in a blue font at lines 447-452 of the manuscript.

We have carefully considered your constructive questions and provided detailed explanations while carefully revising the manuscript so that readers can understand our intentions.

Thank you again for reviewing our manuscript and acknowledging the relevant work we have done.

In equation 28, the left hand side of equation should be BCELoss not "N".

Response: First, we apologize for making this mistake in equation 28. After your severe suggestions and reminders, we carefully modified equation 28 accordingly. The left hand side of equation 28 should be “BCELoss” instead of “N”.

In equation 28, what does "n" represent. It should be replaced with N (as N represent number of answer categories.)

Response: First, we apologize for making this mistake in equation 28. After your severe suggestions and reminders, we double-checked Equation 28, where "n" should be "N", which we have carefully modified.

Reviewer #2: This article improves the performance of the VQA task by an introduction of a gating mechanism in intra- and inter-modal interactive learning, along with a multi-modal adaptive gated mechanism during multi-modal fusion. To better control the contribution of each modal feature to the predicted answer, the authors propose utilizing the self-attention gated unit and the self-guided attention gated unit. Experiment results on the VQA2.0 and GQA benchmark datasets show the relevence of the proposed model. In general, the contribution of this work is significant. I recommend this work to be published.

Response: We are very grateful that you took the pains to review our manuscript and were able to acknowledge our relevant work in the manuscript.

---

## [Decision Letter · Decision Letter 1]

7 Jun 2023

Multi-modal Adaptive Gated Mechanism for Visual Question Answering

PONE-D-23-05435R1

Dear Dr. Zhang,

We’re pleased to inform you that your manuscript has been judged scientifically suitable for publication and will be formally accepted for publication once it meets all outstanding technical requirements.

Kind regards,

Muhammad Bilal

Academic Editor

PLOS ONE

Additional Editor Comments (optional):

The comments of reviewers can be found at the end of this email. The authors are suggested to carefully review the paper in the light of reviewer comments such as typing mistake "Chapter" word used in the paper as raised by the reviewer.

Reviewers' comments:

Reviewer's Responses to Questions

**Comments to the Author**

1. If the authors have adequately addressed your comments raised in a previous round of review and you feel that this manuscript is now acceptable for publication, you may indicate that here to bypass the “Comments to the Author” section, enter your conflict of interest statement in the “Confidential to Editor” section, and submit your "Accept" recommendation.

Reviewer #2: All comments have been addressed

Reviewer #3: All comments have been addressed

2. Is the manuscript technically sound, and do the data support the conclusions?

Reviewer #2: Yes

Reviewer #3: Yes

3. Has the statistical analysis been performed appropriately and rigorously? 

Reviewer #2: Yes

Reviewer #3: Yes

4. Have the authors made all data underlying the findings in their manuscript fully available?

Reviewer #2: Yes

Reviewer #3: Yes

5. Is the manuscript presented in an intelligible fashion and written in standard English?

Reviewer #2: Yes

Reviewer #3: Yes

6. Review Comments to the Author

Reviewer #2: The comments of the reviewers have been addressed adequately. At this stage, the contribution of this work is clearly stated and significant. The authors proposed a novel multimodal adaptive gated mechanism model. The model proves to be efficient experimentally. In overall, I am satisfied and recommend this work to be published.

Reviewer #3: The article seems to be derived from the thesis, please replace the word "Chapter" with "Section" :

Section 3 Method: Line 1

Section 4 Experiment: Line 1

7. PLOS authors have the option to publish the peer review history of their article (what does this mean?). If published, this will include your full peer review and any attached files.

Reviewer #2: **Yes: **Thien Nguyen

Reviewer #3: No

---

## [Editor Report · Acceptance letter]

13 Jun 2023

PONE-D-23-05435R1 

Multi-modal Adaptive Gated Mechanism for Visual Question Answering 

Dear Dr. Zhang:

I'm pleased to inform you that your manuscript has been deemed suitable for publication in PLOS ONE. Congratulations! Your manuscript is now with our production department. 

Kind regards, 

on behalf of

Dr. Muhammad Bilal 

Academic Editor

PLOS ONE